# Enhancing Cervical Cancer Screening with 7-Type HPV mRNA E6/E7 Testing on Self-Collected Samples: Multicentric Insights from Mexico

**DOI:** 10.3390/cancers16132485

**Published:** 2024-07-08

**Authors:** Carlos Eduardo Aranda Flores, Bente Marie Falang, Laura Gómez-Laguna, Guillermo Gómez Gutiérrez, Jorge Miguel Ortiz León, Miguel Uribe, Omar Cruz, Sveinung Wergeland Sørbye

**Affiliations:** 1Oncology Service, Hospital General de México “Dr. Eduardo Liceaga”, Ciudad de México 06720, Mexico; carlos.aranda@salud.gob.mx (C.E.A.F.); laura.gomez2021@enah.edu.mx (L.G.-L.); displasias@hotmail.com (G.G.G.); chomplis571020hc0@hotmail.com (J.M.O.L.); 2PreTect AS, 3490 Klokkarstua, Norway; bente.falang@pretect.no; 3Reyna Madre Clinc, Toluca de Lerdo 50120, Mexico; doc.muribeu@gmail.com; 4Colposcopy Clinic “Fundacion Dr. Fernando Cruz Talonia”, Ciudad de México 09440, Mexico; omarcyc@prodigy.net.mx; 5Department of Clinical Pathology, University Hospital of North Norway, 9019 Tromsø, Norway

**Keywords:** cervical cancer, HPV screening, self-sampling, human papillomavirus, HPV mRNA, E6/E7, positive predictive value, negative predictive value, risk stratification

## Abstract

**Simple Summary:**

This study explores an innovative approach to cervical cancer screening by employing a 7-type HPV mRNA E6/E7 test on self-collected samples from women in Mexico. It investigates whether this method can enhance the accuracy of diagnostics and increase acceptance among participants referred for colposcopy and biopsy due to abnormal cytology results (ASC-US+). The potential benefits include reducing the rate of cervical cancer and the need for unnecessary medical procedures, making screening more accessible and acceptable, especially in underserved areas. The findings could significantly influence the future practices of cervical cancer screening.

**Abstract:**

Cervical cancer remains a significant public health issue, particularly in regions with low screening uptake. This study evaluates the effectiveness of self-sampling and the 7-type HPV mRNA E6/E7 test in improving cervical cancer screening outcomes among a referral population in Mexico. A cohort of 418 Mexican women aged 25 to 65, referred for colposcopy and biopsy due to abnormal cytology results (ASC-US+), participated in this study. Self-samples were analyzed using both the 14-type HPV DNA test and the 7-type HPV mRNA E6/E7 test. The study assessed the sensitivity, specificity, positive predictive value (PPV), and the necessity of colposcopies to detect CIN3+ lesions. Participant acceptability of self-sampling was also evaluated through a questionnaire. The 7-type HPV mRNA E6/E7 test demonstrated equivalent sensitivity but significantly higher specificity (77.0%) and PPV for CIN3+ detection compared to the 14-type HPV DNA test (specificity: 45.8%, *p* < 0.001). The use of the HPV mRNA test as a triage tool reduced the number of colposcopies needed per CIN3+ case detected from 16.6 to 7.6 (*p* < 0.001). Self-sampling was highly accepted among participants, with the majority reporting confidence in performing the procedure, minimal discomfort, and willingness to undertake self-sampling at home. Self-sampling combined with the 7-type HPV mRNA E6/E7 testing offers a promising strategy to enhance cervical cancer screening by improving accessibility and ensuring precise diagnostics. Implementing these app roaches could lead to a significant reduction in cervical cancer morbidity and mortality, especially in underserved populations. Future research should focus on the long-term impact of integrating these methods into national screening programs and explore the cost-effectiveness of widespread implementation.

## 1. Introduction

Cervical cancer continues to pose a critical public health issue in Mexico, where it is the second leading cause of cancer-related deaths among women, despite having organized screening programs since the 1970s [1,2]. The inefficacy of these programs has been associated with issues in quality and coverage [3]. Efforts have been made to reorganize the program to improve secondary prevention. Recent advances in our understanding of the molecular pathogenesis of cervical cancer have led to significant improvements in prevention and screening methodologies. Key developments include prophylactic HPV vaccines and the implementation of high-throughput HPV DNA testing, which play crucial roles in reducing the incidence of cervical cancer through early detection and timely intervention [4,5,6]. Furthermore, the adoption of self-sampling for HPV testing represents a groundbreaking approach to expand screening access, particularly for women who are less likely to engage with conventional screening methods. This strategy is reinforced by robust global evidence, prompting the World Health Organization’s 2021 guidelines to favor primary HPV testing over cytology, whether samples are self-collected or taken by providers [7,8,9].

Mexico has been a pioneer in incorporating HPV testing into its screening program, including self-collected samples [10,11]. A significant study published in 2011 by Eduardo Lazcano-Ponce et al. underscored the efficacy of self-collected vaginal samples for HPV testing among Mexican women, highlighting Mexico’s role as possibly the first country to adopt HPV primary testing and cytological triage as a national policy [12,13]. Despite these advancements, participation in screening programs remains low, with only about 30–40% of Mexican women regularly taking part. These programs still predominantly rely on conventional cytology [14,15]. The lack of adherence to current guidelines and the procedures for referring patients with abnormal cytological findings to colposcopy are major hurdles in the effectiveness and timeliness of screening and treatment [16]. These observations underscore the continued need for improved strategies to lower barriers to participation, utilize accurate diagnostics, and ensure prompt diagnosis and treatment. The limitations of cytology as a screening tool point to a necessity for a more specific diagnostic approach, particularly in the triage of low-grade cytological abnormalities, to better balance the benefits and risks associated with cervical cancer screening [17,18,19].

In this context, exploring molecular biomarkers linked to the progression risk of HPV-induced lesions has gained momentum. Notably, the overexpression of HPV E6/E7 mRNA has proven to be a promising biomarker for refining risk stratification [20,21,22,23,24,25,26,27]. The risk of progression to cervical cancer varies significantly across HPV genotypes and is influenced by the persistence of infection. Comprehensive studies have defined the risk profiles for various HPV genotypes, emphasizing the crucial impact of seven high-risk types (HPV 16, 18, 31, 33, 45, 52, and 58), which are responsible for over 90% of cervical cancers globally [28,29,30,31,32,33,34]. In the era following widespread HPV vaccination, limiting HPV testing to these seven types included in the nonavalent vaccine has been proposed as an improved strategy [35].

Building on previous research by Aranda et al., which confirmed the feasibility and acceptability of self-sampling for HPV testing with results comparable to those of clinically collected samples for both HPV DNA and mRNA detection among Mexican women, this study seeks to expand upon those findings [36]. Our prior research was constrained by only having HPV prevalence data without histological confirmation of high-grade lesions. This current multicentric study incorporates histological analysis within a referral population. By focusing on the clinical utility of the 7-type HPV mRNA E6/E7 test in self-collected samples for triaging abnormal cytology results, this research aims to offer important insights for enhancing cervical cancer screening protocols in Mexico.

## 2. Materials and Methods

### 2.1. Study Population and Recruitment

This multicentric study, conducted in 2022, targeted a cohort of 421 women aged 25–65 years who were referred for colposcopy and biopsy services at three distinct medical facilities: Oncology Service, General Hospital of Mexico, Dr. Eduardo Liceaga, (GHMEL); “Reina Madre” Clinic; and Colposcopy Clinic “Fundacion Dr. Fernando Cruz Talonia”. Eligibility criteria were set to include women who received abnormal cytology results (ASC-US+) during primary screening and who had no history of cervical cancer treatment interventions.

Prior to enrollment, each participant was required to provide informed consent. To ensure a comprehensive understanding of each participant’s medical history, detailed medical history forms were filled out, and case record forms were meticulously maintained throughout the study duration. To facilitate the self-sampling process, participants were provided with both verbal explanations and written instructions, complete with illustrations. Additionally, to evaluate the acceptability and feasibility of the self-sampling methodology, participants were asked to complete a questionnaire following the procedure.

### 2.2. Study Protocol and Procedures

The study was designed around a sequential process involving three distinct patient visits: the initial enrollment and self-sampling, a colposcopy examination and biopsy session, and a final visit for histological assessment and therapeutic guidance.

### 2.3. Self-Sampling Procedure (Visit 1)

Participants were provided with the XytoTest self-sampling device (Mel-Mont Medical Inc., Doral, FL, USA) and instructed to follow the manufacturer’s guidelines for sample collection [37]. After self-sampling, each participant handed the device to a clinician, who then placed it in a tube containing 5 mL of PreservCyt solution (Hologic, Inc., Marlborough, MA, USA) to preserve the sample until HPV testing. These samples were anonymized to maintain confidentiality and stored under controlled conditions until analysis. Molecular HPV DNA testing was performed initially, followed by mRNA analysis on the residual sample within six weeks of collection. Post-collection, a clinician performed a visual inspection of the cervical-vaginal area to ensure there were no adverse effects from the sampling process.

### 2.4. Colposcopy and Biopsy (Visit 2)

Participants underwent a colposcopy examination within two to five days following self-sampling, adhering to the ASCCP Colposcopy Standards [38]. The examination aimed to identify areas of potential pre-cancerous changes for targeted biopsy. In instances where the colposcopy did not indicate specific areas of concern, blind biopsies were taken from standardized locations (12, 3, and 9 o’clock positions) to ensure comprehensive evaluation. Histological analysis was conducted by an expert panel comprising pathologists from Mexico and Norway. Findings were categorized into diagnostic groups based on the WHO classification for cervical intraepithelial neoplasia (CIN 1-3) and adenocarcinoma in situ [39].

### 2.5. Follow-Up and Treatment (Visit 3)

Following the histological analysis, participants were scheduled for a follow-up consultation to discuss the results. Treatment plans and follow-up schedules were determined in accordance with the histological findings and aligned with Mexican national guidelines for the management of cervical pathology. 

### 2.6. HPV DNA Testing

The molecular analysis of all collected samples was conducted using the Abbott RealTime HR HPV test (Abbott, Wiesbaden, Germany), adhering strictly to the guidelines provided by the manufacturer [40]. This test uses a multiplex polymerase chain reaction (PCR) technique that can partially genotype HPV 16 and HPV 18. Additionally, it combines the detection results for twelve other high-risk HPV types (31, 33, 35, 39, 45, 51, 52, 56, 58, 59, 66, and 68) into a single pooled result. To ensure the integrity and quality of each sample, the presence of human beta-globin is assessed as an internal control. The determination of HPV positivity is based on a cycle threshold (CT) value set below 32.0, serving as the critical threshold for identifying the presence of HPV DNA in the samples. 

### 2.7. HPV mRNA E6/E7 Testing

For the detection of E6/E7 mRNA expression from the high-risk HPV types 16, 18, 31, 33, 45, 52, and 58, the study employed the CE-IVD certified PreTect HPV-Proofer`7 assay (PreTect AS, Klokkarstua, Norway). This assay leverages real-time nucleic acid sequence-based amplification (NASBA) technology, targeting the full-length transcripts of the E6 and E7 oncogenes. The assay integrates an intrinsic sample control (ISC) aimed at a universally expressed human housekeeping gene, serving to assess specimen quality and identify potential factors that may impede amplification.

The testing protocols strictly adhered to the manufacturer’s guidelines [41]. In the event a sample returned an invalid result due to a negative mRNA intrinsic sample control, a retest was mandated. Samples failing to produce a valid result upon retesting were excluded from further analysis to uphold the study’s integrity and the reliability of its findings.

### 2.8. Study Outcomes

Primary outcome was histologically confirmed CIN3+ diagnosis from the biopsies collected during colposcopy that was triggered by an abnormal cytology result in primary screening. Since CIN2 has greater variability in diagnosis and is more prone to regression, this analysis focused on CIN3+ cases solely. Calculations of sensitivity, specificity, positive predictive value (PPV), and negative predictive value (NPV) were made to evaluate the performance of 14-type HPV DNA and 7-type HPV mRNA testing. 

### 2.9. Statistical Analysis

The analysis of the data was carried out using the Statistical Package for Social Sciences version 29.0. The significance of associations was evaluated through the Chi-square test with a predetermined significance level of *p*-value < 0.05.

### 2.10. Ethical Considerations

The study protocol received approval from the Ethics and Research Committees at the General Hospital of Mexico, under reference number DI/16/111/03/001. All study participants provided written informed consent before their inclusion in the study.

## 3. Results

### 3.1. Study Population Characteristics

The valid study population included 418 Mexican women aged 25 to 65, with an average age of 40.1 years (±10.7 years, standard deviation). These participants were recruited from three different hospitals within Mexico City, reflecting a diverse study cohort (Table 1). Examination of their cervical cytology screening history revealed a bifurcation in their screening practices. Specifically, 52.2% (218 out of 418) of the participants reported adhering to an annual cervical cytology screening regimen, indicating a pattern of regular engagement with screening services. Conversely, 46.6% (195 out of 418) disclosed that they did not participate in annual screenings, highlighting a segment of the population with irregular screening attendance (Table 1). These characteristics are provided as context for our findings; however, the detailed results of the HPV testing and histology have been discussed separately.

### 3.2. HPV Prevalence and Genotype Distribution

Among the 421 participants who provided self-collected samples, three samples (1 HPV DNA and 2 HPV mRNA) returned invalid test results and were excluded from further analysis. Within the valid study population, 95.0% (397 out of 418) exhibited abnormal cytology results (ASC-US+). Specifically, 70.8% (296 out of 418) presented with low-grade squamous intraepithelial lesions (ASC-US/LSIL), and 24.2% (101 out of 418) with high-grade squamous intraepithelial lesions. Notably, the prevalence of CIN3+ lesions within the cohort was 3.6% (15 out of 418), including four cases of invasive cervical cancer.

The overall prevalence of high-risk HPV (hr-HPV+) genotypes was 55.6% (232 out of 418) using the 14-type HPV DNA test and 25.5% (106 out of 418) utilizing the 7-type HPV mRNA test. The detailed genotype prevalence, as identified by both testing methodologies, is summarized in Table 2. HPV 16 emerged as the predominant genotype detected by both assays: 14.4% (60 out of 418) through the 14-type HPV DNA test and 10.6% (44 out of 418) via the 7-type HPV mRNA test. Additionally, HPV 18’s detection rate was 2.9% (12 out of 418) with the 14-type HPV DNA test and 1.2% (5 out of 418) with the 7-type HPV mRNA test. A higher prevalence of other high-risk HPV genotypes (non-16/18) was identified by the 14-type HPV DNA test, with a rate of 38.4% (160 out of 418), compared to 13.7% (57 out of 418) observed with the 7-type HPV mRNA test (Table 2).

### 3.3. Comparative Performance of HPV Testing in Detecting CIN3+ Lesions

The efficacy of the 7-type HPV mRNA E6/E7 test in identifying CIN3+ lesions was assessed against that of the 14-type HPV DNA test. Both tests demonstrated comparably high sensitivity rates for CIN3+ detection, with the 7-type HPV mRNA test achieving a sensitivity of 93.3% (14 out of 15 cases), mirroring the performance of the 14-type HPV DNA test (*p* = 1.000), as outlined in Table 3, Table 4 and Table 5. Notably, the 7-type HPV mRNA test exhibited a significantly higher specificity of 77.0% (308 out of 400) compared to the 14-type HPV DNA test, which had a specificity of 45.8% (184 out of 402) (*p* < 0.001). Furthermore, the positive predictive value (PPV) for CIN3+ detection was higher for the 7-type HPV mRNA test at 13.2% (14 out of 106) compared to 6.0% (14 out of 232) observed with the 14-type HPV DNA test (*p* = 0.042). The negative predictive value (NPV) remained comparably high for both tests, with 99.7% (308 out of 309) for the 7-type HPV mRNA test and 99.5% (184 out of 185) for the 14-type HPV DNA test (*p* = 1.000), as shown in Table 3, Table 4 and Table 5.

### 3.4. Efficiency in Detecting CIN3+ Lesions

The adoption of the 7-type HPV mRNA E6/E7 test as a triage tool markedly improved the efficiency of identifying CIN3+ lesions, significantly reducing the number of colposcopies needed to detect one case of CIN3+. For each case of CIN3+ detected, the 14-type HPV DNA test necessitated an average of 16.6 colposcopies. In contrast, the 7-type HPV mRNA test required significantly fewer, averaging just 7.6 colposcopies (*p* < 0.001), effectively reducing the number of colposcopies by half (Table 6).

### 3.5. Acceptability of Self-Sampling

The study also evaluated participant perspectives on self-sampling through a questionnaire focused on four critical aspects: confidence in self-administration, discomfort level, ease of procedure, and willingness for future self-sampling. The feedback was very positive: a significant majority (93.1%) of participants felt confident in their ability to perform self-sampling. Most women (74.1%) reported experiencing minimal discomfort during the process. The procedure was deemed easy by 85.0% of respondents, and a high willingness to engage in self-sampling at home was expressed by 91.4% of participants. These results are presented in Table 7, which includes responses from all 421 participants who consented and completed the questionnaire, regardless of the validity of their HPV test results.

## 4. Discussion

This study marks the first comparative evaluation of the 7-type HPV mRNA test against the 14-type DNA test using self-collected cervicovaginal samples for detecting CIN3+ among a Mexican population referred for colposcopy due to abnormal cytology results. It highlights that while DNA tests detect the presence of the virus, mRNA tests identify transcriptionally active infections, thereby increasing specificity. This is significant as the overexpression of mRNA for the viral oncogenes E6 and E7 is acknowledged as a direct driver of cervical cancer progression [42,43,44,45].

### 4.1. HPV Detection, Genotype-Specific Risk, and CIN3+ Prevalence

For this study cohort, our analysis revealed the prevalence of high-risk HPV to be 55.6% using the DNA test and 25.5% with the mRNA test. The prevalence of CIN3+ was 3.6% (15 out of 418), which included four cases of invasive cervical cancer. Despite the different positivity rates, both tests identified the same 14 out of 15 women with CIN3+, including all four cervical cancer cases. The observed hr-HPV prevalence corresponds with that of Oyervides-Muñoz et al., who reported a 60.5% high-risk HPV DNA positivity rate among 294 Mexican patients with gynecological abnormalities referred for colposcopy. Notably, HPV 16 was the predominant genotype detected in both studies, accounting for 25% (60 out of 232) of infections in our study compared to 33.7% (60 out of 178) in the study by Oyervides-Muñoz et al. [10]. The results presented confirm earlier findings that DNA tests typically yield positivity rates approximately twice as high as those of mRNA tests when triaging women with low-grade cytological lesions and CIN3+ outcomes [21,25].

The narrower range of HPV types detected by the mRNA test affects the results. However, the increase in mRNA expression with lesion severity enhances the consistency of type-specific results between mRNA and DNA tests in high-risk populations, making the difference between the two tests less substantial compared to low-risk populations [46]. Recent studies have highlighted significant differences between DNA and mRNA detection rates per genotype in low-risk populations undergoing primary HPV screening, demonstrating a heightened genotype-specific risk associated with mRNA. A case in point is the Norwegian real-life data reported by Sørbye et al., which found hr-HPV in 5.6% (990/17,684) of women participating in the national cervical cancer screening program. Specifically, HPV DNA type 16 was detected in 130 women, resulting in 45 CIN2+ cases, whereas HPV mRNA type 16 was identified in 73 women, leading to 39 CIN2+ cases, with estimated genotype-specific risks of 34.6% for DNA 16 and 53.4% for mRNA 16 [47]. Although our study concentrated on CIN3+ with a relatively small number of cases, we observed similar trends: mRNA demonstrated 1.5–2 times higher risk estimates than DNA, even within a referral population.

### 4.2. Test Performance Evaluation against CIN2+ or CIN3+

Histology is often regarded as the gold standard for evaluating screening strategies and diagnostic tests. Although many studies assess outcomes against CIN2+, it is crucial to recognize that CIN2 diagnoses are subject to considerable variability, and their suitability as an intervention threshold is frequently debated [48]. A systematic review and meta-analysis by Tainio et al. reported that approximately half of CIN2 lesions regress within two years, and just under a fifth may progress to more severe stages. Notably, this study found higher rates of regression and lower rates of progression in women under 30 years old [49].

The US Preventive Services Task Force advises that the management of screen-positive results should focus on the immediate risk of CIN3+ rather than CIN2+ owing to the high likelihood of spontaneous regression in CIN2 cases. This guidance is based on risk-based principles that aim to balance the benefits of detecting and treating CIN3+ to prevent invasive cancer against potential harms. These harms include the risks and costs associated with unnecessary treatments and colposcopies, which may not yield additional clinical benefit if the lesions are unlikely to progress to cancer [50]. In this study, evaluating the test performances specifically against CIN3+ significantly enhanced the sensitivity of the mRNA test compared to its performance against CIN2+. This distinction is essential for programs that continue to use CIN2+ as a threshold for clinical intervention.

### 4.3. Clinical Implications

The clinical implications of our findings are profound, particularly due to the higher specificity of the mRNA test. This significantly reduces the number of unnecessary colposcopies required per detected CIN3+ case, a crucial measure of screening efficacy. In our study, the 7-type mRNA test resulted in significantly fewer colposcopies compared to the 14-type DNA test (7.6 vs. 16.6), demonstrating its enhanced precision in identifying clinically relevant lesions. Both tests exhibited high sensitivity for CIN3+ detection, successfully identifying 14 out of 15 cases. However, the mRNA test not only showed higher specificity (77% vs. 45.8%) but also a positive predictive value (PPV) twice that of the DNA test. These findings corroborate other research that underscores the importance of targeting a limited number of HPV genotypes in the prevention of precancerous lesions and cervical cancer [21,23,25,51,52,53,54].

### 4.4. CIN3+ Risk in Triage-Negative Women

The varied response to infection across different HPV genotypes underscores the need for targeted screening strategies that can adapt to the changing landscape of HPV prevalence, especially in the post-vaccination era. However, the potential for abnormalities in patients who test negative in low-grade cytology triage has raised concerns, prompting calls for longitudinal studies, particularly regarding mRNA tests that target a limited number of HPV types.

Recently, Rad et al. reported a 12-year follow-up of 9582 women who were mRNA-tested for five HPV types (16, 18, 31, 33, and 45) with concurrent normal cytology results, showing a low long-term risk of CIN3+ (1.1%) for mRNA-negative women [55]. Additionally, Rad et al. compared the triage performance of a 13-type HPV DNA test to that of a 5-type HPV mRNA test among 4115 women with ASC-US/LSIL screening results, finding a 6-year CIN3+ risk of 2.8% for mRNA-negative women and 1.4% for DNA-negative women [56]. These findings address previously mentioned concerns, demonstrating that the future risk of abnormalities in mRNA-negative women is sufficiently low, thereby supporting the reliability of mRNA testing in triage settings.

### 4.5. How Many Types to Screen for

The World Health Organization (WHO) categorizes 14 HPV types as carcinogenic, although evidence suggests type 66 should be removed from future tests [57]. The International Agency for Research on Cancer (IARC) classifies 12 types as carcinogenic to humans based on prevalence, not potency [58]. Studies show that while genotype distribution varies by geographic area, the types found in cervical cancer tissue are consistent globally [59]. A Mexican study of 60,135 women from 20 states identified the most common genotypes in precancerous lesions as 16, 31, 33, 58, 18, and 45, with types 56, 66, and 51 absent in precancerous lesions and cervical cancer cases [28].

Research conducted in Sweden revealed that 85.3% of cervical cancers detected through screening were linked to HPV types 16, 18, 31, 33, 45, or 52. Adding the eight additional HPV types that most screening tests cover increased the detected prevalence only marginally, by 1.5% [31]. Meanwhile, a study from Norway found these additional eight types in merely 1.4% of the diagnosed cancers, with the primary types of HPV (16, 18, 31, 33, and 45) accounting for 93.0% (66 out of 71) of the cases [60].

In light of these findings, standard HPV testing includes 14 different HPV types. However, recent evidence, including from Nygård et al., suggests that in the era following widespread vaccination, focusing HPV screenings on the types included in the nonavalent vaccine could optimize test performance [35]. This more targeted approach underscores the potential for the 7-type HPV mRNA test to offer more specific and efficient screening in both vaccinated and unvaccinated populations. The 7-type HPV mRNA test, by focusing on fewer types but those most linked to cervical cancer, could serve as a complementary tool to vaccination strategies. It can bridge the gap until the vaccinated population reaches an age where the risk of cervical cancer is significantly reduced, providing an effective interim solution as well as a long-term screening strategy.

### 4.6. Barriers to Screening and Implementation of Self-Sampling

Integrating innovative approaches into global health strategies is crucial to overcome screening barriers and accelerate the elimination of cervical cancer, aligning with the WHO’s ‘90-70-90’ strategy [61]. The high acceptability and effectiveness of self-sampling reported should encourage health policymakers to reconsider current screening protocols to enhance coverage and reduce the burden of cervical cancer, particularly in underserved populations [62,63,64,65]. Recently, the FDA’s landmark approval of self-sampling for HPV testing has been announced. This pivotal decision is expected to broaden access to screening, lower barriers to testing, and offer more individuals the chance for early detection, treatment, and improved survival outcomes [66].

Several studies have investigated the reasons for low attendance to screening and identified potential barriers throughout the entire process [67,68,69,70]. Women in low- and middle-income countries often face challenges related to accessible sample collection, diagnosis, and timely treatment. However, sociocultural and healthcare system barriers exist worldwide, and the Mexican healthcare system is no exception. 

In 1999, Lazcano-Ponce and colleagues critically evaluated the efficacy of the Mexican screening program, noting the persistently high cervical cancer mortality rate (16 per 100,000 women) 15 years after the screening program’s implementation. They attributed the program’s ineffectiveness mainly to issues related to quality and coverage across all program components. Factors such as poor sample collection quality, low sensitivity of cytology readings, limited coverage, and inadequate knowledge among women about the importance of Pap smears were highlighted. To improve prevention, they suggested a reorganization that included increasing coverage, improving sample collection control, better interpretation of Pap smears, ensuring treatment, and enhancing follow-up [3,71]. Twelve years later, Lazcano-Ponce et al. continued their work, focusing on the initial steps of this chain—specifically, sample collection by the women themselves and the use of accurate molecular testing. Data demonstrated that self-collection for HPV testing was highly acceptable and four times more sensitive than clinician-collected cytology for detecting CIN2+, including cancers [12]. This underscores the potential of self-collection to increase screening attendance and reduce loss to follow-up among those who test positive.

In Latin America, the ESTAMPA study (EStudio multicéntrico de TAMizaje y triaje de cáncer de cuello uterino con pruebas del virus del PApiloma humano) revealed that despite global evidence and WHO recommendations to implement primary HPV testing, cytology remains the main method for cervical cancer screening [72]. Ramírez et al. presented the results of the largest cervical cancer screening study to date in Latin America. They stated, ‘Our results are consistent with previous evidence demonstrating that cytology does not perform accurately to detect precancer and cancer, whereas HPV testing is highly sensitive. The limited accuracy of cytology, which remains the main screening method in Latin American countries, along with limited coverage and inadequate follow-up of abnormalities, likely explains the persistently high burden of disease in the region’. The discussion also addressed the limitations of sensitive primary screening due to its low specificity, resulting in over-referrals and potential overtreatment [17].

The integration of the 7-type HPV mRNA E6/E7 test as a triage tool significantly enhanced the precision of identifying CIN3+ lesions in the current study, as shown by the reduced number of colposcopies required per case detected. This finding supports the growing evidence that mRNA testing, by focusing on the expression of viral oncogenes, correlates more directly with the presence of precancerous and cancerous lesions compared to DNA testing alone. Our results also underscored the high acceptability of self-sampling among participants, indicating its potential to increase screening coverage, particularly among under-screened populations.

### 4.7. Considerations and Future Directions

Our findings advocate for a paradigm shift towards the integration of mRNA testing and self-sampling in cervical cancer screening programs. Future research should explore the implementation of these methods in broader population-based screening settings, evaluating their impact on participation rates, cost-effectiveness, and, ultimately, on the incidence and mortality of cervical cancer. Additionally, longitudinal studies are necessary to understand the long-term outcomes of women triaged using mRNA tests, including the progression or regression of precancerous lesions and the impact on treatment protocols.

The high acceptability of self-sampling suggests its potential as a game-changer in reaching under-screened or inaccessible populations. Leveraging digital health technologies to facilitate self-sampling and remote consultation could further enhance the scalability and effectiveness of screening programs. Furthermore, exploring the integration of self-sampling with point-of-care HPV testing could offer immediate insights into risk stratification, empowering timely decision-making and personalized care pathways.

In summary, the implementation of the 7-type HPV mRNA E6/E7 test as a triage tool, alongside the promotion of self-sampling, presents a promising approach to enhance the efficacy and accessibility of cervical cancer screening. By prioritizing high-risk infections and offering a non-invasive sampling method, these strategies hold the potential to significantly reduce the global burden of cervical cancer, particularly in regions with low screening uptake. Future efforts should focus on overcoming these limitations and expanding our understanding of the long-term benefits derived from integrating these innovative methods into routine screening programs.

### 4.8. Strengths and Limitations

A key strength of our study lies in its real-world applicability, demonstrated through the recruitment of participants from multiple healthcare settings across Mexico City. The use of both HPV DNA and mRNA tests has provided a comprehensive understanding of their comparative efficacy in a screening context. The meticulous follow-up for histological confirmation of CIN3+ lesions adds to the robustness of our findings, providing a strong basis for the recommended shift in screening practices.

However, our study is not without limitations. Excluding participants without a history of cervical cancer treatment may limit the generalizability of our findings across all women eligible for cervical cancer screening. Additionally, the study’s reliance on self-reported data for previous screening history could introduce recall bias, affecting the accuracy of historical screening data. Our study focused exclusively on outcomes related to CIN3+, which presents a limitation due to the absence of data on CIN2. This omission restricts the applicability of our findings to programs that prioritize CIN2 as the clinical action point. Further, the study’s design—focusing on a cross-sectional analysis—precludes the assessment of longitudinal outcomes, including the natural history of HPV infections detected by either test. Additionally, while our study highlights the clinical advantages of the 7-type HPV mRNA E6/E7 test, it does not fully explore the economic implications of its broader implementation. Future research is suggested to thoroughly explore the cost-effectiveness of the 7-type HPV mRNA E6/E7 test, providing a clearer understanding of the full scope of implications for adopting this test in clinical practice. Further, another limitation of our study was the practice of immediate preservation of the self-collected samples in a methanol-based preservative solution to ensure RNA stability. This approach, while effective for maintaining sample integrity for RNA testing, can only be administered within healthcare institutions due to the hazardous nature of methanol, making it unsuitable for at-home use. As a result, our methodology does not fully leverage the potential benefits of self-sampling to increase screening coverage. Additionally, while dry sampling could enhance accessibility and coverage, the time delay in transporting samples to the laboratory for preservation might adversely affect RNA integrity, potentially compromising test accuracy. Therefore, further stability testing of dry sampling, particularly under conditions where temperatures may exceed ambient levels, is essential to optimize the implementation of self-sampling and mRNA testing in widespread practice. Addressing these challenges will be crucial for maximizing the effectiveness and accessibility of cervical cancer screening programs.

## 5. Conclusions

This study underscores the feasibility and high acceptability of self-sampling as an innovative approach for HPV testing within a referral population. Notably, the 7-type HPV mRNA E6/E7 test outperformed the 14-type HPV DNA test in terms of specificity and positive predictive value (PPV) for the detection of CIN3+ lesions. The strategic use of the HPV mRNA test as a triage tool markedly decreased the necessity for colposcopies. This could facilitate a clearer distinction between women who require immediate intervention and those suitable for routine follow-up.

By integrating self-sampling with advanced molecular diagnostic techniques, our findings pave the way for transformative advancements in cervical cancer screening programs. This approach not only enhances screening accessibility but might also ensure precise and timely identification of high-risk cases, thereby optimizing patient management and contributing significantly to cervical cancer prevention efforts.

Future initiatives should focus on broadening the application of these methodologies in diverse settings to fully assess their potential in reducing cervical cancer incidence and mortality on a global scale.

## Figures and Tables

**Table 1 cancers-16-02485-t001:** Study population characteristics.

Age	Mean	SD
Years	40.1	±10.7
**Recruitment source**	**n**	**%**
Oncology Service, GHMEL	191	45.7
Reina Madre Clinic	66	15.8
Colposcopy Clinic	161	38.5
**Annual cervical cytology examination**	**n**	**%**
Yes	218	52.2
No	195	46.6
Not reported	5	1.2
**Age at first sexual intercourse**	**n**	**%**
<18	222	53.1
>18	193	46.2
Not reported	3	0.7

**Table 2 cancers-16-02485-t002:** HPV genotype prevalence by 14-type HPV DNA test and 7-type HPV mRNA test.

n = 418	14-Type HPV DNA	7-Type HPV mRNA
	n (%)	n (%)
HPV 16	60 (14.4)	44 (10.6)
HPV 18 (non-16)	12 (2.9)	5 (1.2)
HPV other * (non-16/18)	160 (38.4)	57 (13.7)
Any hr-HPV+	232 (55.6)	106 (25.5)

* Other HPV DNA (31, 33, 35, 39, 45, 51, 52, 56, 58, 59, 66, and 68). Other HPV mRNA (31, 33, 45, 52, and 58).

**Table 3 cancers-16-02485-t003:** Contingency table for 14-type HPV DNA test vs. CIN3+ outcomes.

Outcome	Histology Positive	Histology Negative	Total
Test Positive	14	218	232
Test Negative	1	184	185
Total	15	402	417

**Table 4 cancers-16-02485-t004:** Contingency table for 7-type HPV mRNA test vs. CIN3+ outcomes.

Outcome	Histology Positive	Histology Negative	Total
Test Positive	14	92	106
Test Negative	1	308	309
Total	15	400	415

**Table 5 cancers-16-02485-t005:** Sensitivity and specificity for CIN3+ for the 14-type HPV DNA test and the 7-type HPV mRNA test.

Triage Test	Sensitivity % (n)	Specificity % (n)	PPV % (n)	NPV % (n)
14-type HPV DNA	93.3 (14/15)	45.8 (184/402)	6.0 (14/232)	99.5 (184/185)
7-type HPV mRNA	93.3 (14/15)	77.0 (308/400)	13.2 (14/106)	99.7 (308/309)

**Table 6 cancers-16-02485-t006:** Contingency table for number of colposcopies per CIN3+ detected.

Triage Test Total	Women Screened	Test Positive	CIN3+ Detected	Colposcopies Per CIN3+
14-type HPV DNA Test	417	232	14	16.6
7-type HPV mRNA Test	415	106	14	7.6

**Table 7 cancers-16-02485-t007:** Self-sampling acceptability questionnaire responses.

Aspects	Number of Responses (n)	Percentage (%)
1. Level of discomfort		
1	312	74.1
2	56	13.3
3	22	5.2
4	13	3.1
5	8	1.9
6	3	0.7
7	3	0.7
8	1	0.2
(Total responses)	421	100.0
2. Level of difficulty		
1	358	85.0
2	41	9.7
3	14	3.3
4	5	1.2
5	0	0.0
6	1	0.2
7	1	0.2
8	1	0.2
(Total responses)	421	100.0
3. Would you perform self-sampling at home?		
Yes	385	91.4
No	36	8.6
(Total responses)	421	100.0
4. Do you feel confident taking the sample?		
Yes	392	93.1
No	29	6.9
(Total responses)	421	100.0

Respondents scored items on an 8-point Likert scale, ranging from 1 (“no discomfort”) to 8 (unbearable discomfort”).

## Data Availability

The data presented in this study are available on request from the corresponding author due to privacy restrictions related to patient data.

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
