# Peer review of "Enhancing Cervical Cancer Screening with 7-Type HPV mRNA E6/E7 Testing on Self-Collected Samples: Multicentric Insights from Mexico"

_cancers, 2024, doi:10.3390/cancers16132485_

Round 1

Reviewer 1 Report

Comments and Suggestions for Authors

Very good study that yields interesting results on triaging of low-grade cytology lesions.

However, more information on self-sampling would be appreciated. Where did women take their samples, at home or in doctor's office? Who put the sample on the Preservcyt medium and how long did the sample stay out of the medium. This is crucial to translate the protocol to real life. Self- sampling is usually considered not appropriate for RNA tests due to quick degradation of RNA, and manipulation of Preservcyt or other preserving media should not be offered to patients.

Author Response

Reviewer 1, Comment: "Very good study that yields interesting results on triaging of low-grade cytology lesions. However, more information on self-sampling would be appreciated. Where did women take their samples, at home or in doctor's office? Who put the sample on the Preservcyt medium and how long did the sample stay out of the medium. This is crucial to translate the protocol to real life. Self-sampling is usually considered not appropriate for RNA tests due to quick degradation of RNA, and manipulation of Preservcyt or other preserving media should not be offered to patients."

Our response: We appreciate your insightful comments and the opportunity to provide further details on the self-sampling procedure used in our study, which we realize is crucial for replicating the protocol in real-life settings.

In our study, self-sampling was conducted at the hospital, coinciding with the day when cervical cytology was performed by clinicians. Once the sample was collected, it was handed to the clinician, who placed the device in a container containing 5 ml of PreservCyt Solution (Hologic, UK). This approach was chosen to ensure immediate and proper handling of the samples, aligning with the best practices for sample integrity, particularly for mRNA stability.

Participants used the XytoTest self-sampling device, which comes in two versions: a dry version for at-home collection, and a wet version pre-filled with PreservCyt solution to be used when sampling is performed at a health care institution. For our study, we utilized the wet version of the XytoTest device, which contains the PreservCyt medium, ensuring that the samples were preserved immediately upon collection. This version allows for the storage of samples for up to six weeks before HPV mRNA testing, providing ample time for logistical handling without compromising the RNA integrity.

Unpublished preliminary data on dry sampling for HPV mRNA testing demonstrate that RNA remains stable for up to 7 days when kept at ambient temperatures below 30°C. Further stability testing of dry sampling, particularly for geographical areas where temperatures might exceed ambient levels, is necessary. It is important to note that the test includes an intrinsic sample control to verify the presence of sufficient sample material and to confirm the integrity of the RNA, ensuring no deterioration.

The immediate processing and the use of the wet XytoTest device with PreservCyt solution are critical components of our study protocol that enhance the feasibility of self-sampling for RNA-based tests in a clinical setting. We believe these measures effectively mitigate the risks associated with RNA degradation and align with the requirements for high-quality RNA samples necessary for accurate HPV mRNA testing.

Our action: To address your feedback and improve the manuscript’s clarity on this aspect, we will enhance the section detailing the self-sampling procedure to include these specifics, ensuring that future researchers and practitioners fully understand the conditions under which the samples were collected and preserved. Additionally, we have included wet versus dry sampling in the limitation section to acknowledge the potential impact of sample preservation methods on the applicability of our findings. We emphasize the need for further stability testing of dry sampling, particularly under conditions where temperatures may exceed ambient levels. This testing is essential to optimize the implementation of self-sampling and mRNA testing in widespread practice.

Reviewer 2 Report

Comments and Suggestions for Authors

Comment to Editors and Authors

This article dealt with priority of mRNA-E6/E7 in 7 types of HPV in patients with SILs (low- and high- grade squamous intraepithelial lesions) and their relative disease comparing to regular HPV-DNA test and cytology screening. This study was a retrospective, exploring, multicentric cohort study, as it were, to provide a new platform for world-wide or national-wide screenings for cervical cancer.

First, the cost-convenient research between this screening and on-going screening has not elucidated, so the possibility to alternativeness could not be explained.

Second, if this new study is super-expensive it would not be able to choice this. Because the world-wide HPV vaccination strategy would be cheaper if done by national-wide political remedy in women’s health (not only women but also men, if SIL guard 9).

The HPV vaccination is must in all people at 9 to12 years old, it is estimated that cervical cancer by HPV would disappear, like smallpox.

I think the discussion about the cost stated above is mandatory.

To authors

You should demonstrate the actual cost per women and compare to on-going screening. It is true that the cost relates to numbers of participants because of commercial discounting, however, the rough estimated expensive should be demonstrated.

Table 3 and 4 should be written as contingency tables in patient’s numbers.

Q1 In table 5, why the total numbers are 421? The participants were 218.

Comments on the Quality of English Language

Author Response

Reviewer 2, Comment 1: This article dealt with priority of mRNA-E6/E7 in 7 types of HPV in patients with SILs (low- and high- grade squamous intraepithelial lesions) and their relative disease comparing to regular HPV-DNA test and cytology screening. This study was a retrospective, exploring, multicentric cohort study, as it were, to provide a new platform for world-wide or national-wide screenings for cervical cancer.

Our response: Thank you for summarizing the objectives and scope of our study. We are glad that the potential impact of our research on global cervical cancer screening practices was well-understood. The primary aim of this multicentric cohort study was to evaluate the clinical utility of mRNA-E6/E7 testing in detecting high-risk HPV infections and to compare its performance with conventional HPV-DNA testing and cytology. The findings underscore the enhanced specificity of the 7-type HPV mRNA test, offering crucial insights that could help refine screening protocols and policies. The use of self-sampling highlighted in the study is particularly significant as it promotes accessible and patient-centered screening strategies, potentially increasing participation rates and overall screening efficacy.

Our action: We believe the reviewer’s comment does not require changes to the manuscript as it acknowledges the significance and methodology of our study accurately.

Reviewer 2, Comment 2: First, the cost-convenient research between this screening and on-going screening has not elucidated, so the possibility to alternativeness could not be explained.

Our comment: The reviewer notes that our manuscript does not include a cost-effectiveness analysis comparing the proposed 7-type HPV mRNA E6/E7 screening with ongoing screening methods, thus limiting the discussion on its potential as an alternative screening method.

Our response: Thank you for pointing out the absence of a cost-effectiveness analysis in our study. We acknowledge this as a limitation in the current scope of our manuscript. The primary focus of our research was to establish the clinical efficacy of the 7-type HPV mRNA E6/E7 test compared to the standard HPV-DNA tests and cytology in terms of sensitivity and specificity.

However, we recognize the importance of cost-effectiveness analysis in the broader application of new screening technologies. Such analyses are crucial for policymaking and the adoption of new technologies in national screening programs. While we have not included this analysis in our current manuscript, we aim to address this in future research. This will involve a detailed assessment of the costs associated with the mRNA testing method compared to traditional methods, factoring in the cost of materials, labor, and the potential reduction in unnecessary procedures due to higher specificity.

Our Action: We propose to add a statement in the Discussion section acknowledging this limitation and suggesting the need for future research to explore the cost-effectiveness of the 7-type HPV mRNA E6/E7 test. This will provide readers with a clearer understanding of the full scope of implications for adopting this test in clinical practice.

Reviewer 2, Comment 3: Second, if this new study is super-expensive it would not be able to choice this. Because the world-wide HPV vaccination strategy would be cheaper if done by national-wide political remedy in women’s health (not only women but also men, if SIL guard 9). The HPV vaccination is must in all people at 9 to12 years old, it is estimated that cervical cancer by HPV would disappear, like smallpox.

Our comment: The reviewer discusses the potential high costs of the new 7-type HPV mRNA E6/E7 test compared to the global strategy of HPV vaccination, emphasizing that widespread HPV vaccination could be a more cost-effective approach to potentially eradicate cervical cancer similarly to smallpox.

Our response: Thank you for your insightful comments regarding the economic and strategic implications of introducing a new screening method alongside global HPV vaccination efforts. We agree that widespread HPV vaccination is the most effective long-term strategy for eradicating cervical cancer and is a global public health priority.

However, our study emphasizes that despite the ongoing vaccination efforts, there remains a critical need for effective screening methods. The 7-type HPV mRNA E6/E7 test (P7) offers several advantages, particularly in terms of specificity for high-risk HPV types associated with cervical cancer. This is crucial for reducing overtreatment and unnecessary follow-ups in both vaccinated and unvaccinated populations.

Moreover, as the benefits of vaccination will take decades to manifest fully—given the age at vaccination and the latency of cervical cancer—the P7 test provides a more immediate, precise screening tool. This makes it especially valuable in the interim period and potentially as a long-term solution to complement vaccination efforts.

The P7 test could serve as a more specific triage tool in vaccinated populations, helping to pinpoint those who still require intervention despite vaccination, thereby optimizing healthcare resources and focusing attention where it is most needed.

Our action: We will enhance our discussion in the manuscript to clarify the role of the P7 test as not only an interim solution but also as a potentially superior screening option that can coexist with vaccination strategies. This will include highlighting how the P7 test can offer more targeted and efficient screening in both vaccinated and unvaccinated populations.

Reviewer 2, Comment 4: You should demonstrate the actual cost per women and compare to on-going screening. It is true that the cost relates to numbers of participants because of commercial discounting, however, the rough estimated expensive should be demonstrated.

Our comment: The reviewer requests a demonstration of the actual cost per woman for the new 7-type HPV mRNA test compared to the ongoing 14-type HPV DNA test screening, suggesting that a rough estimate of the expenses should be included, despite possible commercial discounting.

Our response: We appreciate the reviewer’s emphasis on the importance of providing a cost comparison between the 7-type HPV mRNA E6/E7 test and the existing 14-type HPV DNA test. Understanding the economic impact of implementing new screening methods is crucial for informed decision-making in healthcare policy.

While our study primarily focuses on the clinical benefits of the 7-type HPV mRNA test, particularly its higher specificity which could potentially reduce the number of unnecessary follow-up procedures such as colposcopies, biopsies, and treatments, we acknowledge the importance of cost considerations in the broader implementation of new screening technologies.

Given the complexities involved in accurate cost assessment, which includes variable pricing, regional cost differences, and the evolving nature of healthcare economic analyses, we feel that a detailed cost comparison at this stage might not accurately reflect the potential economic benefits and could vary significantly by healthcare setting.

Our action: We will address this topic within the 'Limitations and Future Research' section of our manuscript. Here, we will discuss the necessity of further studies focused on the economic evaluation of the 7-type HPV mRNA test compared to traditional screening methods. We will suggest that future research should aim to develop a comprehensive cost-benefit analysis that includes not only the direct costs of testing but also the potential savings from reduced follow-up interventions and the long-term benefits of more accurate screening. This will highlight the need for a nuanced approach to evaluating the economic impact of innovative screening technologies in various healthcare contexts.

Reviewer 2, Comment 5: Table 3 and 4 should be written as contingency tables in patient’s numbers.

Our response: Thank you for your valuable suggestion to represent the data in Tables 3 and 4 as contingency tables. We agree that presenting the data in this format can enhance clarity and facilitate a more straightforward interpretation of the results. We have revised these tables accordingly to display the numbers of patients in a contingency table format, which now includes the total counts of participants tested, the counts of positive and negative results for each test, and the distribution of outcomes (CIN3+ vs. others).

Revised Tables:

Table 3a. Contingency Table for 14-type HPV DNA Test vs. CIN3+ Outcomes

Outcome            Histology Positive            Histology Negative         Total

Test Positive                      14                           218                                         232

Test Negative                    1                             184                                         185

Total                                      15                           402                                         417

Table 3b. Contingency Table for 7-type HPV mRNA Test vs. CIN3+ Outcomes

Outcome            Histology Positive            Histology Negative         Total

Test Positive                      14                           92                                           106

Test Negative                    1                             308                                         309

Total                                      15                           400                                         415

Table 3c. Sensitivity and specificity for CIN3+ for the 14-type HPV DNA test and the 7-type HPV mRNA test

Triage test

Sensitivity

% (n)

Specificity

% (n)

PPV

 % (n)

NPV

 % (n)

14-type HPV DNA

93.3 (14/15)

45.8 (184/402)

6.0 (14/232)

99.5 (184/185)

7-type HPV mRNA

93.3 (14/15)

77.0 (308/400)

13.2 (14/106)

99.7 (308/309)

Table 4. Contingency Table for Number of Colposcopies per CIN3+ Detected

Triage Test         Total Women Screened Test Positive      CIN3+ Detected Colposcopies per CIN3+

14-type HPV DNA Test                   417                         232                         14                           16.6

7-type HPV mRNA Test                  415                         106                         14                           7.6

These revisions provide a clear breakdown of test results against CIN3+ outcomes and help visualize the efficiency and diagnostic performance of each testing method. We appreciate your guidance in improving the presentation of our study results.

Reviewer 2, Comment 6: Q1 In table 5, why the total numbers are 421? The participants were 218.

Our response: Thank you for your inquiry regarding the total number of participants reported in Table 5 of our manuscript. To clarify, our study initially included 421 women who were eligible and consented to participate in the study at the time of recruitment. These participants were selected based on their referral for colposcopy and biopsy due to abnormal cytology results.

Of these, three participants' samples (1 HPV DNA and 2 HPV mRNA) returned invalid test results, reducing the number of valid cases for the analysis of test sensitivity and specificity to 418. However, all 421 participants completed the self-sampling acceptability questionnaire, which is the data represented in Table 5.

This discrepancy between the number of valid test results and the number of questionnaire responses is due to the inclusion of all initial participants in the survey portion of the study, regardless of the validity of their test results. We apologize for any confusion this may have caused and appreciate the opportunity to clarify this aspect of our study design. In response to your comment, we will ensure that this explanation is clear in the manuscript to avoid any misunderstanding.

Reviewer 3 Report

Comments and Suggestions for Authors

Dear authors,

thanks for this relevant study in the  clinical implementation of HPV mRNA self testing, this ma have relevant policy implication, even if larger studies are needed to validate results.

Introduction is fine

Methodology is fine and accurate, I agree with the overall study design.

Results are well explained and conclusions are supported by evidences.

However regarding discussion about widening the number of HPV genotypes in cervical screening, having a full high risk HPV test is beneficial to Maximise the cervical screening impact.

Furthermore, and this should be stressed, the inclusion of other high risk genotypes may increase the efficacy of collateral screening, like in vaginal intraepithelial neoplasia and invasive vaginal carcinoma (please refer to https://doi.org/10.1002/jmv.29474)

Thank you, I am looking forward to receiving the revised version of the manuscript

Comments on the Quality of English Language

Minor

Author Response

Reviewer 3, Comment 1: thanks for this relevant study in the clinical implementation of HPV mRNA self testing, this ma have relevant policy implication, even if larger studies are needed to validate results.

Introduction is fine

Methodology is fine and accurate, I agree with the overall study design.

Results are well explained and conclusions are supported by evidences.

Our reponse: We sincerely appreciate your positive feedback and valuable endorsement of our study on the clinical implementation of HPV mRNA self-testing. We are delighted to hear that you found our methodology to be accurate and the results well explained, and that you agree with our overall study design.

We acknowledge your point regarding the necessity for larger studies to further validate our results. We agree that expanding this research to include a broader demographic and larger sample size would be beneficial to enhance the generalizability of our findings and potentially influence health policy decisions regarding cervical cancer screening more effectively.

Your insights have been instrumental in affirming the direction and implications of our research, and we are motivated to pursue further studies to continue building on this foundational work. Thank you for your constructive and encouraging comments.

Reviewer 3, Comment 2: However regarding discussion about widening the number of HPV genotypes in cervical screening, having a full high risk HPV test is beneficial to Maximise the cervical screening impact.

Our response: Thank you for your comments regarding the scope of HPV genotypes included in our screening approach. We appreciate your perspective on the potential benefits of a broader HPV test that includes all high-risk types to maximize the impact of cervical cancer screening.

Our study specifically focused on a targeted approach using the 7-type HPV mRNA E6/E7 test, which was selected based on its clinical relevance and its ability to detect the most oncogenic HPV types directly implicated in the progression to high-grade lesions and cervical cancer. This assay includes HPV types 16, 18, 31, 33, 45, 52, and 58, which are known to be responsible for over 90% of cervical cancer cases globally.

The rationale for this narrower focus is twofold:

Increased Specificity: By targeting only the most carcinogenic HPV types, the mRNA test enhances specificity for detecting cervical precancers and cancers, thereby reducing the number of unnecessary follow-ups and treatments that could result from broader screening tests. This is particularly relevant in settings where resources for follow-up are limited.

Cost-effectiveness: Although a broader HPV DNA test might capture a greater number of high-risk types, our approach aligns with evidence suggesting that focusing on the major oncogenic types provides a balanced trade-off between cost, specificity, and overall public health impact. Studies indicate that adding additional types beyond the primary oncogenic types contributes minimally to cancer detection rates but may significantly increase the cost and complexity of screening programs.

In the context of current HPV vaccination strategies, which cover several of the high-risk types included in our assay, the relevance of focusing on these types further supports the practical utility and effectiveness of our targeted approach.

Moreover, recent data suggest that with widespread vaccination, the prevalence of diseases caused by the types covered in the nonavalent vaccine is expected to decrease, which may alter the landscape of required testing in the future.

We agree with the need for continual evaluation of screening strategies as vaccination coverage increases and as we gain more understanding of the HPV type distribution in vaccinated populations.

Thank you once again for your insightful feedback, which highlights the dynamic nature of HPV screening in the context of evolving public health strategies.

Reviewer 3, Comment 3: Furthermore, and this should be stressed, the inclusion of other high risk genotypes may increase the efficacy of collateral screening, like in vaginal intraepithelial neoplasia and invasive vaginal carcinoma (please refer to https://doi.org/10.1002/jmv.29474)

Our response: Thank you for your insightful comment on the potential benefits of expanding the range of HPV genotypes included in cervical screening to address collateral screening impacts, such as vaginal intraepithelial neoplasia (VaIN) and invasive vaginal carcinoma.

We acknowledge that the inclusion of a broader spectrum of high-risk HPV genotypes might indeed increase the detectability of other HPV-associated conditions. However, the primary objective of our study was to assess the efficacy and cost-effectiveness of a more focused HPV mRNA test in detecting cervical intraepithelial neoplasia (CIN) and cervical cancer.

The reference you provided indeed discusses the distribution of HPV genotypes in VaIN2/3 and underscores the dominance of HPV 16, which is included in both the 7-type HPV mRNA test and the 9-valent HPV vaccine covered in our study. Importantly, our chosen 7-type test targets the HPV types most implicated in cervical carcinogenesis, which includes the majority of types responsible for serious lesions like VaIN3.

Moreover, while vaginal cancers and higher-grade VaIN are serious concerns, they are relatively rare compared to cervical cancer and thus are not typically the primary target of cervical cancer screening programs. The inclusion of additional genotypes to specifically screen for these rarer conditions would necessitate a different balance of cost, specificity, and potential over-treatment, which might not be justified in broader population screening programs primarily aimed at cervical cancer prevention.

Nevertheless, we agree that this is an area worthy of further research, especially in populations with high incidences of vaginal cancers or where cervical screening is already highly optimized. Future studies could explore the benefits of expanded HPV genotype screening in these contexts.

Thank you once again for raising this important point, which highlights the complexity of designing HPV screening strategies that balance broad protection with cost-effectiveness and clinical utility.

Round 2

Reviewer 2 Report

Comments and Suggestions for Authors

I accept you revised version.

Reviewer 3 Report

Comments and Suggestions for Authors

The paper has been improved but the authors did not add the requested reference

Comments on the Quality of English Language

Minor